# Multipronged Programmatic Strategy for Preventing Secondary Fracture and Facilitating Functional Recovery in Older Patients after Hip Fractures: Our Experience in Taipei Municipal Wanfang Hospital

**DOI:** 10.3390/medicina58070875

**Published:** 2022-06-30

**Authors:** Yu-Pin Chen, Wei-Chun Chang, Tsai-Wei Wen, Pei-Chun Chien, Shu-Wei Huang, Yi-Jie Kuo

**Affiliations:** 1Department of Orthopedics, Wan Fang Hospital, Taipei Medical University, Taipei 116, Taiwan; 99231@w.tmu.edu.tw (Y.-P.C.); 99292@w.tmu.edu.tw (W.-C.C.); 108030@w.tmu.edu.tw (P.-C.C.); 111022@w.tmu.edu.tw (S.-W.H.); 2Department of Orthopedics, School of Medicine, College of Medicine, Taipei Medical University, Taipei 110, Taiwan; 3Department of Nursing, Wan Fang Hospital, Taipei Medical University, Taipei 116, Taiwan; 93409@w.tmu.edu.tw

**Keywords:** hip fracture, fracture liaison service, outcomes, stratified care

## Abstract

*Background and Objectives*: The study assessed the effectiveness of a fracture liaison service (FLS) after 1 year of implementation in improving the outcomes of hip fracture surgery in older adult patients at Taipei Municipal Wanfang Hospital. *Materials and Methods*: The Wanfang hospital’s FLS program was implemented using a multipronged programmatic strategy. The aims were to encourage the screening and treatment of osteoporosis and sarcopenia, to take a stratified care approach for patients with a high risk of poor postoperative outcomes, and to offer home visits for the assessment of environmental hazards of falling, and to improve the patient’s adherence to osteoporosis treatment. The clinical data of 117 and 110 patients before and after FLS commencement, respectively, were collected from a local hip fracture registry; the data were analyzed to determine the outcomes 1 year after hip fracture surgery in terms of refracture, mortality, and activities of daily living. *Results*: The implementation of our FLS significantly increased the osteoporosis treatment rate after hip fracture surgery from 22.8% to 72.3%, significantly decreased the 1-year refracture rate from 11.8% to 4.9%, non-significantly decreased 1-year mortality from 17.9% to 11.8%, and improved functional outcomes 1 year after hip fracture surgery. *Conclusions*: Implementation of our FLS using the multipronged programmatic strategy effectively improved the outcomes and care quality after hip fracture surgery in the older adult population, offering a successful example as a valuable reference for establishing FLS to improve the outcomes in vulnerable older adults.

## 1. Introduction

Osteoporosis-induced fragility fractures, predominantly in the hip and spine, are a grave health concern in older adult patients. Of all osteoporosis-related fractures, hip fracture is the most debilitating injury and is a growing public health concern in the context of an aging population [1,2]. In Asia, the number of cases of hip fracture is estimated to increase from 1,124,060 in 2018 to 2,563,488 in 2050, contributing to a corresponding increase in the direct cost of hip fracture treatment from USD 9.5 to USD 15 billion [3]. In addition, the prognosis of older adults after hip fractures is poor. The 1-year mortality rate associated with a geriatric hip fracture ranges from 14.0% to 18.1% [4,5,6], but it can be as high as 36% 1 year after surgery [7]. In our previous study, up to 33.9% of the 281 older adult patients with hip fractures exhibited severe dependence and required additional care at the 1-year follow-up [8]. Moreover, patients with hip fractures were five times more likely to experience a hip refracture within 1 year [9]. Thus, public health measures and a robust treatment protocol for hip fracture are crucial.

To provide improved care for fragility fractures, the International Osteoporosis Foundation (IOF) advocated the Capture the Fracture campaign in 2013 to raise awareness regarding secondary fracture prevention [10]. Fracture liaison services (FLSs) have been recommended as a coordinator-based best practice program for the care of patients with fragility fractures [10]. This involves systematic investigation and risk assessment to reduce the refracture risk and improve survival [11]. For patients with hip fractures, these FLS programs have been demonstrated to be cost-effective [12] and to reduce both secondary fracture incidence and mortality rates [13].

FLS has been implemented in our hospital since July 1, 2019. However, in contrast to the majority of the FLS programs executed worldwide that focus on osteoporosis detection and treatment, we implemented a multipronged programmatic strategy including encouraging postoperative osteoporosis and sarcopenia screening and treatment, taking a stratified care approach for patients with a high risk of poor postoperative outcomes, and offering home visits for the assessment of environmental hazards of falling and assistance. The program also focused on adherence to taking prescribed antiosteoporosis medications (AOMs), and the aim was to reduce the refracture risk in older adult patients with hip fractures. By evaluating the refracture risk, 1-year mortality rates, and changes in 1-year-postoperative activity of daily living (ADL), the present study assessed the effectiveness of our FLS program after its implementation for 1 year in comparison with the results before FLS implementation.

## 2. Materials and Methods

### 2.1. Program Description

Taipei Municipal Wanfang Hospital is a medical center in Taipei, Taiwan, performing hip fracture surgery in approximately 180 patients annually. Since 1 December 2017, all older adult patients who were scheduled for hip fracture repair were prospectively followed-up and registered in a local hip fracture registry. The patients were included in the registry if they: were at least 60 years old; had a hip fracture, namely femoral neck fracture (FNF) and peritrochanteric fracture (PTF); and were scheduled for surgery, namely hemiarthroplasty or internal fixation with intramedullary nailing by using in situ cannulated screws or dynamic hip screws. Patients were excluded if they were to undergo hip surgery for a reason other than primary hip fracture, including osteoarthritis, trauma, tumor, infection, and avascular necrosis of the femoral heads. Data on demographic characteristics, pre-fracture ADL, and postoperative outcomes were collected for all patients. All the registered patients were routinely followed-up by telephone to gather data on ADL and postoperative complications, including refracture events and mortality, 1 year after hip fracture surgery. From 1 July 2019, Taipei Municipal Wanfang Hospital began to implement FLS for older adult patients with hip fractures as a physician-led intervention. It involves a multipronged programmatic strategy that investigates and treats osteoporosis, provides shared care by physiotherapists and geriatricians for personalized post-surgery rehabilitation programs and comorbidities management, respectively, and offers a stratified care approach for patients with a high risk of poor postoperative outcomes (Figure 1). This FLS of Wanfang hospital was awarded the Gold Level standard as part of the Capture the Fracture program by the IOF in 2021.

In our FLS program, all patients with hip fractures were encouraged to receive operation within 48 h after admission. After operation, all patients had to complete full workups for osteoporosis by dual-energy X-ray absorptiometry (DXA). In addition, all participants were encouraged to undergo sarcopenia screening during admission where handgrip strength was measured and muscle mass was assessed through DXA. Patients with hip fractures were then classified as those with low or high refracture risk based on the presence of concomitant sarcopenia, comorbidities or their bone mineral density. High-risk patients were defined as having concomitant sarcopenia, T-score ≤ −3.0, or more than three comorbidities. After hip fracture surgery, physiotherapists arranged personalized rehabilitation programs for all patients. However, for high-risk patients, geriatricians were also consulted for comorbidity management, duplicate medication screening, and nutrition support during admission.

Once the patient was discharged from the hospital, a prescription of AOMs and calcium and vitamin D supplements within 3 months after surgery was encouraged for all patients with a confirmed diagnosis of osteoporosis through DXA. Three, six, and nine months after surgery, all patients with hip fractures were assessed by an FLS coordinating nurse for AOM use. Patients who failed to continue AOMs after hip fracture surgery and those who were classified as the high-risk group were encouraged to receive home visits by our FLS team members (including orthopedic surgeons and nurses) within 1 year after hip fracture surgery, once consent for home visits was obtained from these patients. During the home visits, we screened and educated the patient on environmental fall hazards at their place of residence, assisted patients who had stopped treatment to adhere to prescribed AOMs, and instructed patients on home-based exercise (Figure 2a,b). After the home visits, the visited patients were contacted by telephone 1 month later for follow-up and to record the changes made in the living place to remove the hazards. Patients who stopped osteoporosis treatment before the home visits were also followed-up after the visits to determine whether they returned to the clinic for AOM treatment.

Because the local hip fracture registry had been collecting patients’ data since 1 December 2017, whereas our FLS program began after 1 July 2019, patients with hip fractures who were scheduled for surgery at Taipei Municipal Wanfang Hospital were thus divided into two groups: a pre-FLS control group and post-FLS intervention group. This study compared the 1-year outcomes of hip fracture surgery, that is, refracture, mortality, and post-fracture ADL, between the pre-FLS and post-FLS groups by using the data extracted from the local hip fracture registry. The entire protocol for the local hip fracture registry and home visits project for the patients were approved by the Ethics Committee of Taipei Medical University, and ethical approval was registered under the serial numbers TMU-JIRB N201709053 and TMU-JIRB N201912066. More specifically, each participant gave their written informed consent to participate in this study. All participants also consented to the publication of their data.

### 2.2. Data Collection

Data on the following basic characteristics were collected: age; sex; body mass index (BMI); Charlson Comorbidity Index (CCI); fracture type; namely FNF or PTF; surgical method; namely joint replacement or internal fixation; surgical time; and blood loss. In addition, surgical delay from admission and results from preoperative serum tests, including those on hemoglobin, creatinine, sodium, and potassium levels were collected. The primary outcomes for comparison between groups included refracture rates (namely all types of osteoporotic fractures including hip, vertebral, radial, and humeral fractures) and mortality rates at the 1-year follow-up. The secondary outcome was post-fracture ADL after 1 year.

### 2.3. Key Performance Indicators for FLS

In our FLS program, we defined several patient-level key performance indicators (KPIs) for FLS to guide quality improvement with reference to the guidance from the IOF Capture the Fracture Campaign [14]. Our KPIs were based on the patient receiving (1) osteoporosis assessment with DXA within 12 weeks after surgery; (2) sarcopenia screening; (3) postsurgery physiotherapy consultation; (4) AOM use (indicated by initiation of treatment, prescription within 3 months after surgery, and continuing AOMs for 1 year after fracture surgery); (5) nutritional supplements (specifically, calcium or vitamin D); and (6) home visits. Data regarding the KPI completion rate were also collected and compared between the pre- and post-FLS groups.

### 2.4. Environmental Evaluation in Home Visits

The environmental evaluations were made by FLS team members by using an environmental checklist during home visits. In our protocol, indoor environmental variables were assessed in two dimensions: (1) environmental fall hazards and (2) environmental protections against falling. Each indoor environmental hazard variable was specifically measured using a dichotomous response (“no” and “yes”) to whether (1) inadequate light (Figure 2e) or (2) other tripping hazards (e.g., cluttered pathways, unsecured rugs, and scattered electrical cords) (Figure 2d) were present. Each indoor environmental protective variable was also assessed using a dichotomous response (“no” and “yes”) to two items: (1) antislip rubber mats in the bathroom and (2) grab bars on the path and in the bathroom (Figure 2c).

### 2.5. Definition of Sarcopenia

Sarcopenia was diagnosed if the patient had low muscle mass and low handgrip strength, as recommended by the Asian Working Group for Sarcopenia (AWGS) [15]. The handgrip strength was measured using a Jamar hydraulic dynamometer (Sammons Preston, Bolingbrook, IL, USA). Handgrip strengths of <28 and <18 kg for men and women, respectively, were regarded as low, based on the thresholds recommended by the AWGS [15]. Muscle mass was represented by the appendicular skeletal muscle mass index (ASMI), which was calculated using DXA. Muscle masses of 7 and 5.4 kg/m^2^ for men and women, respectively, were regarded as low, based on the thresholds recommended by the AWGS [15].

### 2.6. Instruments for Functional Outcomes

The Barthel index (BI), for 10 variables with scores ranging from 0 to 100, is an ordinal scale used for measuring ADL performance and mobility [16]. A higher number is associated with a greater likelihood of being able to live at home independently after being discharged from the hospital. According to the proposed guideline, a BI score of <60 indicates “severe to total” dependency. The BI can be used to accurately assess functional recovery in patients who undergo hemiarthroplasty after FNF [17].

### 2.7. Statistics

All statistical analyses were conducted using IBM SPSS Statistics software version 22 (Armonk, NY, USA). Categorical variables are presented in terms of frequency (percentage) and were compared using the chi-square test and Fisher’s exact test. Continuous variables are presented in terms of the mean ± standard deviation and compared using the Wilcoxon two-sample test and Student’s *t* test.

## 3. Results

### 3.1. Demographic Data

From 1 December 2017 to 30 June 2019 (namely before FLS implementation), data on the basic characteristics of 110 patients undergoing hip fracture repair at Taipei Municipal Wanfang Hospital were collected from the local hip fracture registry; these patients formed the pre-FLS group. Data on the basic characteristics of another 117 patients with hip fractures with complete 1-year follow-up from 1 July 2019 to 30 June 2020 (i.e., after FLS implementation at the Wanfang hospital) were collected; these patients formed the post-FLS group. The basic characteristics of the patients from the pre-FLS and post-FLS groups are presented in Table 1 for comparison. All parameters, including age, sex, BMI, fracture type, CCI, preoperative serum tests, surgical methods, and surgical delay, did not significantly differ between the pre-FLS and post-FLS groups. In addition, no difference in pre-fracture ADL was observed between the patients in the pre-FLS and post-FLS groups.

### 3.2. KPIs for Quality Control in FLS

Table 2 presents statistics on our specific KPIs for quality control in the pre-FLS and post-FLS groups. Patients in the post-FLS group had a higher KPI (all KPIs) completion rate than those in the pre-FLS group; notably, the post-FLS group exhibited significant improvements in the adherence to post-surgery physiotherapy consultation, use of AOMs, consumption of nutritional supplements, and receiving of home visits. In total, 72.3% of patients in the post-FLS group received AOMs, and the majority (62.3%) of them received denosumab as the treatment drug. One year after having a hip fracture, 53% of patients in the post-FLS group continued AOM treatment, but the AOM treatment rate decreased to 14.5% in the pre-FLS group. In addition, 55.6% and 56.4% of the patients in the post-FLS group received nutritional supplements and home visits within 1 year after hip fracture surgery, respectively.

### 3.3. Findings on Home Visits and Changes Thereafter

Table 3 shows a total of 66 patients in the post-FLS group successfully received home visits at a mean of 8.26 months after hip fracture surgery. As for the assessment of indoor environmental hazards, 24.2% and 72.7% of the patients were found to have inadequate light and tripping hazards, respectively, in their place of residence. In addition, only 33.3% and 15.2% of the patients had antislip rubber mats and grab bars, respectively, as environmental protections against falling. However, after home visits by the FL team members, 42.6% of these patients made changes in their environment to prevent falls. Among 30 patients who had stopped AOM use before the home visits, 16 (53.3%) patients successfully returned to a clinic for AOM treatment after the home visits.

### 3.4. Primary and Secondary Outcomes

As indicated in Table 4, the 1-year mortality rate after hip fracture surgery was lower in the post-FLS group than in the pre-FLS group (11.8% versus 17.9%) but not significantly so. However, the refracture rate within the first year after surgery was significantly lower in the post-FLS group than in the pre-FLS group (4.9% versus 11.8%, *p* = 0.048). As for the secondary outcome at the 1-year follow-up, patients in the post-FLS group had a significantly higher ADL score (75.61 ± 30.67) than patients in the pre-FLS group (*p* = 0.018).

## 4. Discussion

Before the implementation of our FLS, the AOM treatment rate after hip fracture surgery was only 22.8%. However, after FLS was implemented, efforts were made to promote osteoporosis screening and treatment and home visits were offered to ensure that patients who had stopped AOM use start using it again, leading to an increase in the AOM treatment rate after hip fracture surgery to 72.3%. Moreover, using a stratified care approach for patients with a high risk of poor postoperative outcomes—including provision of shared care through physiotherapists and geriatricians, as well as indoor environmental assessments by home visits—we successfully decreased the refracture rate from 11.8% before FLS to 4.9% after FLS. Furthermore, the 1-year mortality rate effectively decreased from 17.9% in the pre-FLS group to 11.8% in the post-FLS group. Patients in the post-FLS group also presented with higher ADL 1 year after hip fracture surgery than those in the pre-FLS group. This study demonstrated the effectiveness of our FLS program wherein a multipronged programmatic strategy is used for reducing 1-year refracture and mortality rates and facilitating functional outcomes after hip fracture surgery in the older adult population.

Osteoporosis treatment is a key factor affecting hip fracture outcomes in the older adult population. However, missed diagnosis and the undertreatment of osteoporotic fractures following the first osteoporotic fracture is common and now regarded as a critical clinical concern, and greater effort from both healthcare systems and individual clinicians is required [18]. In the Asia-gap study that surveyed women at postmenopause from seven Asian countries, although 70% patients with hip fractures were aware of the osteoporosis risk, only 25% were assessed for bone mineral density and 30% received AOMs as osteoporosis treatment [19]. The FLS program, which is characteristic of multidisciplinary care allowing for systematic coordination between healthcare professionals, is an effective method for improving investigation, detection, and treatment of osteoporosis following index osteoporotic fracture [10]. A recent study on 724 older adult patients with hip fractures in one medical center in Spain reported that the osteoporosis treatment rate can be increased from 12.3% to 74.9% after FLS implementation [20]. In that study, patients treated with AOMs during FLS implementation had a lower mortality rate than those managed without AOMs before FLS implementation (20.2% versus 25.8%), although FLS implementation seemed not to affect the refracture risk between the pre-FLS and post-FLS groups (4.6% and 3.6%, respectively) [20]. Another study analyzing 75 hip fracture patients under FLS with one-year follow-up in one medical center in Thailand reported that the osteoporotic medication treatment rates increased from 40.8% to 80%, resulting in a significant decline on refracture rate from 30% to 0% [21]. However, the one-year mortality rate was not significantly changed (9.2% and 10.7% in pre- and post-FLS groups, respectively) [21]. Meanwhile, in a study in a teaching hospital in Italy, implementation of FLS in 210 geriatric hip fracture patients was reported to increase the osteoporosis treatment rate from 17.2 to 48.5%, successfully reducing the one-year mortality rate from 15.7% to 12.7% [22]. Moreover, evidence from a recent meta-analysis has also shown that the FLS program can significantly improve the osteoporosis treatment rate, resulting in effectively improved outcomes in terms of reducing future fractures as well as morbidity and mortality [11].

In the present study, our FLS program was found to effectively increase the osteoporosis treatment rate after hip fracture surgery, which not only significantly decreased the refracture rate for all osteoporotic fractures but also improved the mortality rate 1 year after hip fracture surgery in the older adult population. Although our results act in concert with the results of previous reports on the effectiveness of FLS [23], we are convinced that the multipronged programmatic strategy aiming at promoting osteoporosis screening and treatment, increasing patient’s adherence to AOM through home visits, and using a stratified care approach for patients with a high risk of poor postoperative outcomes has also played a crucial role contributing to the positive outcomes in this study. Regarding refracture risk, good compliance to osteoporosis treatment is necessary for fracture risk reduction, with increasing benefit observed with higher compliance [24]. However, because the older adult patients after hip fractures are at a great risk of losing some degree of motility after surgery [8,25], return to clinics for regular AOM treatment may be a difficult task, which may result in poor compliance to AOM treatment after hip fracture surgery. A multicenter study in a high-level intervention FLS reported that the first-year persistence rates for AOM use was only 66.4% after the initiation of osteoporosis treatment [26]. After FLS implementation, among our 74 patients who received initiating treatment with AOMs, 62 (83.8%) patients continued AOM use until 1 year after hip fracture surgery. The high compliance rate in this study may not only attribute to the treatment choice of long-lasting AOMs (the majority of patients (62.3%) in the post-FLS group received denosumab, which was prescribed for once every 6 months), but also result from the efforts of our home visits to recall the patients who had stopped osteoporosis treatment back to the clinics for AOM prescriptions.

In addition to the high treatment rate and compliance to AOM use, a stratified care approach for patients with a high risk of poor postoperative outcomes is also a critical step in our FLS program. Equipped with the knowledge of prognostic factors, clinicians can adopt a stratified care approach by prioritizing older adult patients with hip fractures who are at a high risk of poor functional outcomes or high mortality for intensive care [27]. Considering the findings of our previous study that patients who have hip fractures with baseline sarcopenia [28], a low T-score [28], and high comorbidity [25] are prone to a poor postoperative function and high mortality after hip fracture surgery, we used these three prognostic factors to classify patients with hip fractures as a high-risk group. A Cochrane review for fall prevention reported that several combination exercises led to an approximately 30% reduction in the incidence of falls and home environment adjustment achieved approximately 20% reduction [29]. Therefore, after hip fracture surgery, high-risk patients were obliged to be concomitantly cared for by physiotherapists and geriatricians through personalized rehabilitation programs, nutrition supports, and professional management of comorbidities. After discharge from the hospital, high-risk patients were also encouraged to receive home visits so that indoor environmental hazards of falling could be identified and necessary home environment adjustments could be suggested. Interestingly, during our home visits, we found up to 72.7% patients had tripping hazards in their living place. Nevertheless, through our efforts, 42.6% patients who had environmental hazards successfully changed their environment to prevent falls. Compared with the 8.3% refracture rate within 1 year after the index hip fracture from a large-scale retrospective cohort study [30], our stratified care approach for high-risk patients reinforced the protections against refracture for patients with hip fractures, thereby effectively reducing the 1-year refracture rate to 4.9% after FLS implementation. Moreover, our multipronged programmatic strategy was also demonstrated to be effective in facilitating functional recovery after hip fracture surgery, which may also be attributable to the reduced refracture rate and personalized rehabilitation programs as well as nutritional support for patients with concomitant sarcopenia.

This study has some limitations. First, our FLS program was initiated from July 2019; therefore, we compared the outcomes of patients before and after FLS implementation based on a retrospective analysis of our local hip fracture registry. However, the comparison between the pre-FLS and post-FLS groups was based on a different historical follow-up period rather than on the outcomes from two intervention arms at the same time period. Owing to the potential improvements in surgical techniques and the quality of patient care with time, superior outcomes in the post-FLS group may also be affected by other potential confounding factors and therefore be biased. However, because all patients’ data were extracted from a local registry with high-quality follow-up (the loss follow-up rate for patients 1 year after hip fracture surgery was only 9.5%), our findings likely reflect the efficacy of our FLS program. Second, although rehabilitation programs and nutritional support are essential for patients with hip fractures, especially the high-risk group, the rehabilitation protocol and nutrient regimens cannot be standardized for each patient. Personalized rehabilitation and nutrition support were inevitable and may therefore cause uncontrolled bias. Third, the follow-up period was limited to 1 year, and this may be too short for us to determine the long-term effectiveness of FLS in patients with hip fractures. Finally, the representativeness of our sample was limited by its small size. All participants were recruited from the same institution and might not represent the older adult population undergoing hip fracture surgery throughout Taiwan. Whether our FLS program can be replicated in other institutions to have similar outcomes remains to be clarified. Even with these limitations, we shared our own experience using the multipronged programmatic strategy to improve the care quality after hip fracture surgery in the older adult population, offering a successful example for enhancing and closing the gaps in osteoporosis hip fracture care at Taipei Municipal Wanfang Hospital.

## 5. Conclusions

Our FLS program, which was designed to encourage the screening and treatment of osteoporosis and sarcopenia, to take a stratified care approach for patients with a high risk of poor postoperative outcomes, and to offer home visits for the assessment of environmental hazards of falling, and to improve the patient’s adherence to osteoporosis treatment, was proven to successfully reduced the 1-year refracture rate and facilitated functional recovery after hip fracture surgery in our older adult sample. The experience of our multipronged programmatic strategy for care after hip fracture surgery in the older adult population is anticipated to serve as a valuable reference for establishing FLS to improve the outcomes of vulnerable older adult individuals with hip fractures.

## Figures and Tables

**Figure 1 medicina-58-00875-f001:**
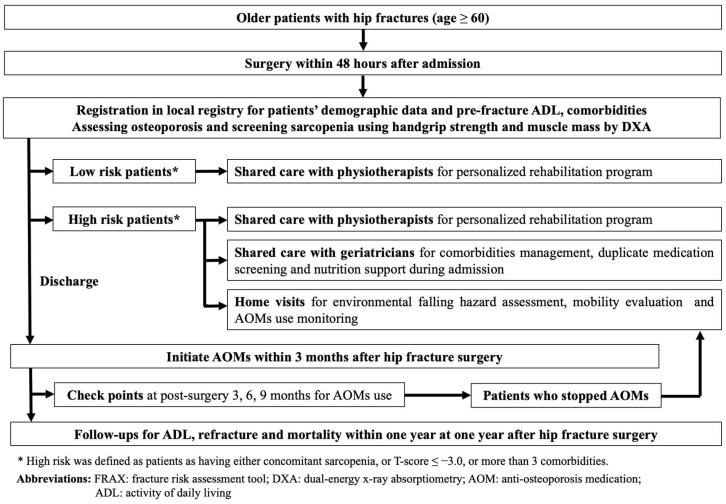
Fracture Liaison Service in Taipei Municipal Wanfang Hospital.

**Figure 2 medicina-58-00875-f002:**
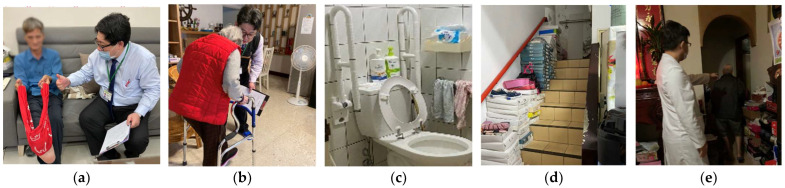
Home visits for patients with hip fractures. (**a**,**b**) Instructing patients about home-based exercise; (**c**) protective device in the living environment, grab bars in the bathroom; (**d**) hazard in the living environment, obstacles on pathway; (**e**) environmental hazard, inadequate light.

**Table 1 medicina-58-00875-t001:** Basic characteristics of patients in the pre-FLS and post-FLS groups.

Variable	Pre-FLS (*n* = 110)	Post-FLS (*n* = 117)	*p* Value
Age	82.98 ± 8.20	80.67 ± 9.76	0.071
Gender			0.66
Female	81 (73.6%)	83 (70.9%)	
Male	27 (26.4%)	34 (29.1%)	
BMI	22.31 ± 3.74	21.58 ± 3.46	0.063
Fracture type			0.349
Femoral neck fracture	66 (60.0%)	62 (53.0%)	
Peritrochanteric fracture	44 (40.0%)	55 (47.0%)	
Lesion side			0.233
Left	52 (47.3%)	65 (55.6%)	
Right	58 (52.7%)	52 (44.4%)	
CCI	5.05 ± 1.74	4.87 ± 1.80	0.475
Preoperative serum tests			
Hemoglobin (g/dL)	12.05 ±1.78	12.18 ± 1.74	0.924
Creatinine (mg/dL)	1.36 ±1.70	1.07 ± 0.96	0.477
Sodium (mmol/L)	137.01 ±3.77	136.62 ± 4.29	0.475
Potassium (mmol/L)	3.94 ±0.51	3.90 ± 0.46	0.617
Surgical methods			0.173
Joint replacement	48 (43.6%)	40 (34.2%)	
Internal fixation	62 (56.4%)	77 (65.8%)	
Surgical delay from admission			0.982
Within 24 h	66 (60%)	70 (59.8%)	
24–48 h	32 (29.1%)	35 (29.9%)	
>48 h	12 (10.9%)	12 (10.2%)	
Surgical time (h)	71.61 ± 26.23	85.92 ± 54.62	0.073
Surgical blood loss	102.00 ± 91.95	106.39 ± 110.84	0.851
Pre-fracture ADL	82.36 ± 24.23	83.25 ± 23.78	0.959

**Table 2 medicina-58-00875-t002:** Comparison of KPIs between the pre-FLS and post-FLS groups.

KPIs	Pre-FLS (*n* = 110)	Post-FLS (*n* = 117)	*p* Value
Assessment with DXA within 12 weeks after surgery	92 (83.6%)	102 (90.9%)	0.449
T-score	−3.93 ± 1.10	−3.90 ± 1.12	0.84
Sarcopenia screening			
Handgrip strength (kg)	14.52 ± 7.93	14.42 ± 12.50	0.382
Muscle mass assessment with DXA	80 (72.7%)	96 (82.1%)	0.093
Muscle mass (ASMI, kg/m^2^)	5.67 ± 1.04	5.70 ± 1.12	0.84
Diagnosis of sarcopenia	42 (53.2%)	49 (51.3%)	0.873
Post-surgery physiotherapy consultation	32 (29.1%)	117(100%)	0.000
AOM use			
Initiating treatment with AOMs	32 (22.8%)	74 (72.3%)	0.000
Prescription within 3 months after surgery	24/32 (75%)	67/74 (90.5%)	0.071
Denosumab	10 (31.3%)	46 (62.3%)	
Bisphosphonate	11 (34.3%)	19 (25.7%)	
Selective estrogen-receptor modulators	7 (21.9%)	2 (2.7%)	
Teriparatide	4 (12.5%)	7 (9.5%)	
Continuing AOMs for 1 year after fracture	16 (14.5%)	62 (53.0%)	
Nutrition supplements (i.e., calcium or vitamin D)	16 (14.5%)	65 (55.6%)	0.000
Receiving home visits	0	66 (56.4%)	0.000

**Table 3 medicina-58-00875-t003:** Findings on home visits and changes thereafter.

Home Visits after Hip Fracture Surgery	*n* = 66
Mean follow-up time after surgery (months)	8.26 ± 3.40
Mean age	79.86 ± 9.52
Environmental evaluation	
Indoor environmental hazards of falling	54 (81.8%)
Inadequate light	16 (24.2%)
Tripping hazards	48 (72.7%)
Indoor environmental protection against falling	26 (39.4%)
Antislip rubber mats in the bathroom	22 (33.3%)
Grab bars on the path and in the bathroom	10 (15.2%)
Changing environmental hazards after visits	23/54 (42.6%)
Stop AOM use before home visits	30
Return to clinics for AOM treatment after home visits	16/30 (53.3%)

**Table 4 medicina-58-00875-t004:** Comparison of 1-year outcomes between the pre-FLS and post-FLS groups.

Outcomes	Pre-FLS (*n* = 110)	Post-FLS (*n* = 117)	*p* Value
Refracture within the first year of surgery	13 (11.8%)	5 (4.9%)	0.048
Mortality within the first year of surgery	17 (17.9%)	12 (11.8%)	0.225
ADL at 1-year follow-up	64.19 ± 34.17	75.61 ± 30.67	0.018

## Data Availability

Due to the sensitive nature of the questions asked in this study, survey respondents were assured that raw data would remain confidential and would not be shared.

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
