# Peer review of "Multipronged Programmatic Strategy for Preventing Secondary Fracture and Facilitating Functional Recovery in Older Patients after Hip Fractures: Our Experience in Taipei Municipal Wanfang Hospital"

_medicina, 2022, doi:10.3390/medicina58070875_

Round 1
Reviewer 1 Report
The fractures in the elderly are a big challenge in medicine.
Multipronged Programmatic Strategy for Preventing Secondary Fracture and Fa- 2 cilitating Functional Recovery in Older Patients After Hip Fractures: Our Experi- 3 ence in Taipei Municipal Wanfang Hospital
the topic described is very interesting, excellent organization of the study. it can be a starting point for many hospitals to create protocols to prevent refractures, despite the management criticalities of this purpose.
the text appears well written in its sections, the conclusions appear poor in content, I would describe and deepen the conclusions.
Author Response
Der reviewer: Thank you for giving us the opportunity to revise our submitted manuscript. We tried our best to reply all the reviewer’s comments point-to-point and edit the manuscript accordingly as in attachment. We hoped that the revised version fulfills the requirements of your esteemed journal.
Reviewer 2 Report
the article is well done. need some minor revision for english language.
Author Response
Dear reviewer: Thank you for giving us the opportunity to revise our submitted manuscript. We tried our best to reply your question as in attachment. We hoped that the revised version fulfills your requirements.
Reviewer 3 Report
The manuscript of Chen et al. ‘Multipronged Programmatic Strategy for Preventing Secondary Fracture and Facilitating Functional Recovery in Older Patients After Hip Fractures: Our Experience in Taipei Municipal Wanfang Hospital’ is presenting single center data on implementing a FLS.
The relevance of the presented data is, due to demographic changes and a high increase in frail patients with hip fractures, very high. Strength of the study is the carefully conducted study design and the clear presentation of the data.
Minor Concern:
- The authors clearly state as a limitation, that the presented data is single center derived. The authors are encouraged to add a comparison of their key findings with the data already given in the literature, e.g. in the form of a mini-review added to the discussion. Numerous centers have already assessed the introduction of a FLS in their hospitals and shared their experience. At this point the authors discuss selected articles from other groups, but do not attempt to benchmark their own finding.
Minor revision is needed.
Author Response

(The authors gave the same response as above.)
